# *LhANS-rr1*, *LhDFR,* and *LhMYB114* Regulate Anthocyanin Biosynthesis in Flower Buds of *Lilium* ‘Siberia’

**DOI:** 10.3390/genes14030559

**Published:** 2023-02-23

**Authors:** Shaozhong Fang, Mi Lin, Muhammad Moaaz Ali, Yiping Zheng, Xiaoyan Yi, Shaojuan Wang, Faxing Chen, Zhimin Lin

**Affiliations:** 1Fujian Academy of Agricultural Sciences Biotechnology Institute, Fuzhou 350003, China; 2College of Horticulture, Fujian Agriculture and Forestry University, Fuzhou 350002, China

**Keywords:** anthocyanin, bulb, *Lilium* ‘Siberia’, full-length transcriptomics, MYB transcription factor, VIGS

## Abstract

The bulb formation of *Lilium* is affected by many physiological and biochemical phenomena, including flower bud differentiation, starch and sucrose accumulation, photoperiod, carbon fixation, plant hormone transduction, etc. The transcriptome analysis of flower buds of Lilium hybrid ‘Siberia’ at different maturity stages showed that floral bud formation is associated with the accumulation of anthocyanins. The results of HPLC-MS showed that cyanidin is the major anthocyanin found in *Lilium* ‘Siberia’. Transcriptome KEGG enrichment analysis and qRT-PCR validation showed that two genes related to flavonoid biosynthesis (*LhANS-rr1* and *LhDFR*) were significantly up-regulated. The functional analysis of differential genes revealed that *LhMYB114* was directly related to anthocyanin accumulation among 19 MYB transcription factors. Furthermore, the qRT-PCR results suggested that their expression patterns were very similar at different developmental stages of the lily bulbs. Virus-induced gene silencing (VIGS) revealed that down-regulation of *LhANS-rr1*, *LhDFR*, and *LhMYB114* could directly lead to a decrease in anthocyanin accumulation, turning the purple phenotype into a white color. Moreover, this is the first report to reveal that *LhMYB114* can regulate anthocyanin accumulation at the mature stage of lily bulbs. The accumulation of anthocyanins is an important sign of lily maturity. Therefore, these findings have laid a solid theoretical foundation for further discussion on lily bulb development in the future.

## 1. Introduction

Lilies (*Lilium*, spp.), a group of monocotyledonous ornamental plants, are widely grown for commercial purposes. They significantly contribute to the global ornamental industry and are utilized for commercial bulb and flower production, including outdoor and indoor fresh-cut flowers and potted plants, and for landscaping in private gardens [1,2]. 

Anthocyanins, a type of flavonoid, are secondary metabolites, exist in many horticultural plants, and have multiple biological roles [3], such as growth inhibition of tumors [4], as well as anti-inflammation and -oxidation [5]. Furthermore, anthocyanins act as plant-coloring substances, play a vital role in defense mechanisms, and protect the plants from UV damage [5]. The rate of anthocyanin synthesis in the plant, membrane transport, and utilization or degradation are the main factors to influence the final anthocyanin concentration in ripe fruits [6,7]. Anthocyanin biosynthesis in fruits may be regulated by the activities of their metabolism-related enzymes, such as L-phenylalanine ammonia-lyase (PAL), cinnamate 4-hydrogenase (C4H), 4-coumarate: coenzyme A Ligase (4CL), chalcone synthase (CHS), UPD-3-O- glycosyltransferase (UFGT), and glutathione S-transferase (GST) [8,9,10,11]. In the anthocyanin biosynthesis pathway, the phenylalanine is converted into cinnamic acid by PAL [12,13]. The 4CL and CHS enzymes are derived from polyketone synthases (PKSs) [14] and are involved in the synthesis of naringin chalcone [15]. Likewise, the main function of the UFGT enzyme is to stabilize anthocyanin biosynthesis by attaching the sugar portion to anthocyanin glycogens, and it is a rate-limiting gene involved in anthocyanin biosynthesis [16]. The GST enzyme is very important for anthocyanin transportation in plants, and the results show that it is a key factor in the expression of anthocyanin-based red coloration in bracts [17].

With the cloning of anthocyanin-related transcription factors, anthocyanin biosynthesis in plants has been formally studied [18]. In plants, anthocyanin synthesis is mainly related to three elements, including two *R2R3-MYB* type TF, two basic helix-loop-helix type TF (*bHLH*), and one *WD40* repeat TF [19], and their interaction [20]. Anthocyanins have been widely studied in Arabidopsis, pear, rice, maize, apple, petunia, and lily [21]. The overexpression of *R2R3-MYB* type TFs in Arabidopsis, including *MYB114*, *MYB113*, *MYB90*, and *MYB75*, enhanced anthocyanin accumulation [22]. Furthermore, in an experiment on red-skinned pear, it was revealed that *PyMYB114* could inhibit anthocyanin biosynthesis, which was similar to the function in tobacco and strawberry [23]. Further research suggests that ethylene response factors interacted with *PyMYB114* to regulate anthocyanin biosynthesis in red-skinned pears [24]. The transcriptome data on *Hydrangea macrophylla* revealed that *MYB114* acted as a negative regulation transcription factor in anthocyanin synthesis [25]. A *WD40* repeat gene (*OsTTG1*) regulated anthocyanin biosynthesis in rice [26]. *ZmC1* (*MYB-type C1*) and *ZmR* (*bHLH R*) are two TFs; their up-regulation controlled anthocyanin biosynthesis in maize [27]. In apples, most *MYB* TFs positively regulated anthocyanin biosynthesis, except for *MdMYB16* [28]. *MdMYB114*’s transcription level was increased with the deepening of apple color, and overexpression promoted the production and accumulation of anthocyanin in callus [29]. 

The buildup of anthocyanin in lily varieties causes the difference in floral color [30]. Several genes and TFs are involved in the anthocyanin synthesis of lily. In the *Lilium* cultivar ‘Montreux’, *LhMYB6* and *LhMYB12* were isolated, which interacted with *LhbHLH2* protein and positively regulated anthocyanin biosynthesis [31]. During flower development, *LhMYB12* could directly activate the promoters of chalcone synthase and dihydroflavonol 4-reductase in Asiatic hybrid lily [32]. The new allele of the *LhMYB12*, named *LhMYB12-Lat*, determined the presence or absence of splatters on the tepal of the Asiatic hybrid lily [33]. However, *R3-MYB* TFs usually play a native role in repressing anthocyanin biosynthesis [34]. For example, the transient expression of the *LvMYB1* gene suggests that it inhibits anthocyanin biosynthesis in lily flowers [35]. 

In this study, a *LhMYB114* transcription factor was isolated from Oriental hybrid *Lilium* ‘Siberia’ based on the transcriptome data. Our study suggested that it can effectively regulate the genetic expression of *ANS* and *DFR* via the VIGS system, thereby affecting the accumulation of anthocyanins in the floral bud of the *Lilium* bulb.

## 2. Materials and Methods

### 2.1. Experimental Material

The floral buds of *Lilium* hybrid ‘Siberia’ were sampled at six development stages according to the size of bulb circumference (Figure 1), where anthocyanin accumulation gradually increased from stage C to stage F. We finally focused on the last two stages (Figure 1E,F) for transcriptomics sequencing, named stage E and F.

### 2.2. Anthocyanidin Identification

Samples were more than 5 g. All samples were homogenized in a homogenizer and stored at −18 °C. The samples were mixed with the HCl (0.5% *v*/*v*) and 70% methanol buffer to a volume of 50 mL, and this process was repeated 3 times. Then, the samples were shook for 1 min and extracted for 30 min by ultrasonication. It was hydrolyzed in a boiling water bath for 1 h, and then cooled in constant volume. Standing and taking the supernatant, it was filtered with 0.45 aqueous phase filter membrane, and stored at 4 degrees for test. The injection volume of 20 μL was passed through the chromatographic column C18. The mobile phase A is an aqueous solution containing 1% formic acid, and the mobile phase B is a 100% acetonitrile solution at 0.25 mL/min flow rate. It was detected by a wavelength of 530 nm at LC-QTOF (USA, Agilent 6545B). Delphinidin, cyanidin, petunidin, pelargonin, and mallow pigment were used as an internal standard for quantitation. 

### 2.3. Transcriptomic Sequencing and Differential Gene Expression Analysis

Total RNA was extracted from the samples of two developmental stages (Figure 1E,F) using RNAprep pure Plant Kit (Tiangen, Beijing, China; Code:DP360) according to the manufacturer’s protocol. HiScript^®^ II 1st Strand cDNA Synthesis Kit (+gDNA wiper) (Vazyme, Nanjing, China) was used as reverse transcription kit. The quantity and quality of the cDNA were assessed by using a NanoDrop ND2000 spectrophotometer (Thermo Scientific, Waltham, MA, USA). cDNA library construction and transcriptome sequencing were performed by Novogene Technologies Co., Ltd. (Beijing, China). The quality of the sample libraries was determined by an Agilent 2100 Bioanalyzer and Qubit2.0. Finally, the well-constructed library was sequenced using an Illumina NovaSeq 6000 platform (Beijing Novogene Technology, Beijing, China). The R package DESeq2 (version 1.10.1) was used to identify differential expression analysis of two groups (E&F), and the gene expression values were calculated as reads aligned to the fold change of the normalized (RPKM) expression values by Ballgown (version 2.14.1). FDR was used to judge the threshold of *p* value. DEGs mainly depended on multiples hypotheses, such as the *p*-value threshold, false discovery rate (FDR) (≤0.05) and (log 2 FoldChange ≥ 1), which were considered as the significance of each gene expression difference.

### 2.4. Assay of Quantitative PCR

The real-time quantitative PCR (qRT-PCR) was performed on an ABI Q1 Real-Time PCR System with 10 µL mixture according to the following procedures: 95 °C for 1 min, 40 cycles of 95 °C for 15 s, and 60 °C for 30 s. The primers were designed using the software Primer Premier 6.0 and are listed in Appendix A. *Lilium × formolongi* EF-1a was selected as a reference gene and the relative expression was analyzed with the 2^−∆∆Ct^ method [36]. Three independent biological as well as technical replications were used.

### 2.5. Virus-Induced Gene Silencing in Lilium

Silencing of the targeted gene (*LhMYB114*, *LhANS-rr1*, or *LhDFR*) by VIGS was performed as described earlier [37,38]. According to the experiment protocol, a 200–500 base pair (bp) fragment, specific to the target gene, was designed and cloned into the pTRV2 vector (primers are listed in Appendix A) using a One-step Cloning Kit (Vazyme, Nanjing, China), which was named 35S:LhMYB115, 35S:LhANS-rr1, and 35S:LhDFR. The blank vector of TRV1, TRV2, and the target genes inserted into TRV2 were transformed individually into the *Agrobacterium* GV3101 strain. Then, monoclonal bacteria were picked up and inoculated in a 100 mL LB medium containing antibiotics, then grown overnight at 28 °C in a shaker. *Agrobacterium* cells were collected and resuspended in infiltration buffer (10 mM MgCl_2_, 200 μM acetosyringone, and 10 mM 2-(N morpholino) ethane sulfonic acid (pH 5.6) to a final OD_600_ of 1.8). Equal volumes of TRV1 and TRV2 (the control), as well as the TRV1 and TRV2 with target genes, were mixed together and kept in the dark for 3–6 h at room temperature before infiltration. The buds were infiltrated using a needleless 1 mL syringe and grew at room temperature for seven days. Then, the flower buds continued vernalization in a refrigerator at 4 °C under dark for 15 days.

### 2.6. Electron Microscopy and Morphological Observation

To evaluate the features of apical meristem structure, the apical meristem from *Lilium* bulbs was cut for scanning electron microscopy (SEM) at 6 developmental stages. The cutting sections were soon fixed in 2.5% glutaraldehyde, and then dehydrated in a series of graded ethanol, as previously described [39]. The samples were observed and photographed with an SEM SU-8010 (Hitachi Ltd., Tokyo, Japan) (Appendix A).

### 2.7. Statistical Analysis

All data are presented as the mean ± standard deviation (SD) of at least three independent replicates. All statistical analyses in this paper were performed with SPSS (version 22.0, USA) software. The significant and extreme differences were confirmed if *p*-values were <0.05 and <0.01, respectively.

## 3. Results

### 3.1. The Chromatographic Analysis of Purple Anthocyanin in Flower Buds

To identify the major anthocyanin compound in *Lilium* flower buds, five types of anthocyanins were used as standard, including delphinidin, cyanidin, petunia, pelargonin, and mallow (Figure 2). Cyanidin was detected as the major anthocyanin compound in *Lilium* flower buds, as suggested by the results of HPLC-MS.

### 3.2. Identification of DEGs Related to Flavonoid Biosynthesis

Based on transcriptome data, putative DEGs from E and F groups identified about 8944 DEGs (Figure 3C), including 4557 transcripts showing up-regulation and 4387 transcripts showing up-regulation. The *Arabidopsis* genome acted as an internal reference. Interestingly, we detected that the sixteen expression genes related to flavonoid biosynthesis were dramatically changed. All the differential expression patterns and gene annotations are shown in the heatmap between E and F groups (Figure 3A). *Lilium* anthocyanins mainly comprised cyanidins; the metabolic pathway is presented in Figure 3B. The quantitative PCR verification showed that the expressions of anthocyanin-related genes were highly consistent with transcriptome analysis, and F3H, LhDFR, and LhANS-rr1 were up-regulated (Figure 3D).

### 3.3. MYB Gene and Cluster Analysis

Based on full-length transcriptomics, 28 kinds of TFs were enriched, such as *C3H*, *MYB-ralated*, *C2H2*, *bHLH*, *GRAS*, *FAR1,* and *AP2*/*EFF-ERF*, etc. Among them, *MYB* and *bHLH* were mainly involved in anthocyanin biosynthesis. The number of *MYBs* reached 33, and *bHLH* reached 54 (Figure 4A). We finally selected the *MYB* transcription factor as the research object, which was the largest one related to anthocyanin synthesis. The results showed that 19 major MYB TFs were related to the differential expression in DEGs, including 8 up-regulated genes and 11 down-regulated genes. Furthermore, only *LhMYB114* was involved in the regulation of anthocyanin biosynthesis (Figure 4B). The result of the evolutionary tree with Arabidopsis showed that it was the homologous genes, with many anthocyanin-biosynthesis-related genes, such as *Atmyb123*, *Atmyb114*, *Atmyb113*, *Atmyb90,* and *Atmyb75,* in Arabidopsis (Figure 4C).

### 3.4. LhMYB114 Gene Cloning and Phylogenetic Analysis

We cloned the *LhMYB114* gene from *Lilium* ‘Siberia’, which includes 214 amino acid sequences and TA insert to pGXT vector for sequencing. The results of structural alignment showed that it belonged to the members of MYB gene family (PLN03212), including R2-R3 repeat domain and the bHLH binding domain (Figure 5). The R2R3-MYB related to anthocyanin pigmentation patterns was highly homologous with *LhMYB114* (Figure 5A).

### 3.5. The Expression Analysis of Three Key Genes Associated with Anthocyanin Metabolism in Flower Buds

We analyzed the expression patterns of three key genes in different development stages, i.e., *LhMYB114*, *LhANS-rr1*, and *LhDFR*. The results showed the expression patterns of the three genes were significantly similar (Figure 6).

### 3.6. Silencing Flavonoid-Biosynthesis-Related Genes Reduced Anthocyanin Accumulation

To determine whether three key genes were essential for anthocyanin accumulation in flower buds, we knocked down *LhMYB114*, *LhANS-rr1*, and *LhDFR* by VIGS. *Agrobacterium* strains of pTRV1 and pTRV2 were mixed in a ratio of 1:1, and buds were infected by syringe and grown for approximately 7 dpi at 25 °C in the dark. Then, we turned it to 7–8 °C ice storage to continue germination for the experiment. The leaf of gene-silenced buds was white around the injection site at 15 dpi, whereas that of control leaf continued to be purple in color (Figure 7A). The PCR amplification results suggested that the target band of RNA1 and RNA2 could be detected in new grown buds of the positive control (empty vector) and gene-silenced plants (pTRV2-*LhMYB114*, *LhANS-rr1*, and *LhDFR* vector), but there were no bands in the negative control (buffer) (Figure 6B). The phenotype and testing results showed that TRV had successfully invaded the buds of *Lilium* and replicated and transferred in vivo. The qRT-PCR results suggested that the expression of three key genes was evidently reduced in gene-silenced buds compared with control buds (empty vector) at 15 dpi (Figure 8A–F). Other genes in the cyanidin pathway were up-regulated, except for the three major genes, which was the opposite of positive regulation (Figure 8G). These results suggested that three key genes played a vital role in anthocyanin accumulation in *Lilium* buds.

## 4. Discussion

The color is an important attribute that determines the quality of flowers [40,41]. The purple color with the development and vernalization in lily buds, which is due to the accumulation of anthocyanins, has become an important target in lily breeding [42]. The anthocyanin accumulation of the bud is not only an indicator of bulb maturity, but also crucial for the development of the bulb and improvements in cold storage resistance [43]. Therefore, it is necessary to analyze the biosynthetic process involved in anthocyanin accumulation in buds, to improve the quality of Lilium buds. In the present study, the results reveal that three genes of *LhANS-rr1*, *LhDFR*, and *LhMYB114* are closely related to anthocyanin biosynthesis in lily buds. Our results (according to VIGS) suggested that three key genes might be closely related to anthocyanin accumulation in floral buds of lilies.

During the process of bulb development, the flower bud of a lily becomes larger with the expansion of bulb size, and anthocyanin accumulation simultaneously occurs. The coordinated expression of genes, which encode the anthocyanin biosynthetic pathway enzymes, usually controls the anthocyanin accumulation through a ternary *MYB-bHLH-WD40* (MBW) transcription complex [44]. Anthocyanin accumulation mainly happens in lily tepals and causes bicolor, bud blush, splatter spots, or raised spots [45,46]. The anthocyanin-accumulation-pathway-related genes in flower buds were significantly enriched in all DEGs. Previous studies showed that a single enzyme-coding structural gene might be involved in anthocyanin biosynthesis, including *C4H*, *4CL*, *CHS*, *CHI*, *UFGT*, *F3H*, *DFR*, and *ANS* genes. Among the unigenes encoding six *CCoAOMT*, *UDPG*, *HCT*, *C4H*, *4CL*, *CHS*, *CHI*, *F3H*, *LhDFR*, *LhANS-rr1*, and *UFGT*, the results suggested that the FPKM values of the three *CCoAOMT*, *LhANS-rr1,* and *LhDFR* unigenes were higher in the purple buds of bulbs than in the white buds, whereas the FPKM values of three *CCoAOMT*, *UDPG*, *HCT*, *C4H*, *4CL*, *CHS*, *CHI*, *F3H*, and *UFGT* were not (Figure 3A). Meanwhile, the qRT PCR results confirmed that the expression of *F3H*, *LhANS-rr1,* and *LhDFR* was significantly up-regulated (Figure 3D). Furthermore, we clone the *LhDFR* and *LhANS-rr1* genes from *Lilium* hybrid ‘Siberia’. The cDNAs in NCBI blast show that both have their alleles in *Lilium speciosum* (*LsDFR-ws* and *LsANS-rr1*), which proved that they were related to anthocyanin biosynthesis [47].

There are three main downstream metabolic pathways of anthocyanins, i.e., cyanidin, pelargonium, and delphinidin, in plants. Though ANS and DFR genes do not specifically regulate anthocyanin biosynthesis, they are the key genes in the cyanidin biosynthesis pathway [48]. Previous studies proved that the structural genes, including DFR, ANS, and 3GT, directly determine the flower color formation in lilies [49]. In our findings, the results of qRT-PCR showed that their expression levels were significantly different at different development stages of *Lilium* bulbs (Figure 6B,C). Further explanation suggested that they might be involved in anthocyanin accumulation in floral lily buds.

*MYB* TFs regulate the phenylpropanoid pathway, including lignin, flavonoids, and other metabolites, in plants [50]. The MBW complex related to anthocyanin biosynthesis is regulated by *MYB* TFs [51]. It is important that MYB is the most abundant TF related to anthocyanins. We analyzed all the DEGs and found that there were 19 major MYB transcription factors. One of them, named *LhMYB114*, was involved in anthocyanin synthesis (Figure 4). The *LhMYB114* sequencing blast results suggest that it is the allele gene of Lilium regale *LrMYB15* for transcription factor R2R3-MYB [52]. The Cluster analysis results also show that *LhMYB114* is homologous with anthocyanin-related genes in *Arabidopsis thaliana*, including *AtMYB123*, *AtMYB5*, *AtMYB114*, *AtMYB113*, *AtMYB90,* and *AtMYB75*. Most R2R3-MYB TFs can promote anthocyanin biosynthesis by up-regulating the expression level in plants [53]. Our results show that the *LhMYB114* expression level increases significantly in the mature stage of lily bulb development (Figure 6A).

The structural genes may be regulated by a single *MYB* transcription factor or the MBW complex [54]. VIGS mediated by *Agrobacterium tumefaciens* is a powerful tool to reveal gene functions in plants, which are difficult to transform, such as lily, potato, etc. [55]. Thus, to prove the regulatory response relationship among them, gene transient silencing is carried out by virus-induced gene silencing (VIGS). The pTRV1 and pTRV2 gene vector-infiltrated flower buds suggested an obvious white phenotype compared with empty vector (pTRV1 and pTRV2) at 15 dpi, including 35S:*LhMYB114*, 35S:*LhANS-rr1*, 35S:*LhDFR*, 35S:*LhMYB114*+35S:*LhANS-rr1*, 35S:*LhMYB114*+35S:*LhDFR*, and 35S:*LhMYB114*+35S:*LhANS-rr1*+*LhDFR* (Figure 8A). The fading phenotype of *LhMYB114* and *LhANS-rr1* was more typical than that of *LhDFR* after VIGS experiments (Figure 7A). At the same time, the qRT-PCR results from six groups showed that the gene expression levels were significantly down-regulated in the new growth buds (Figure 8A–F). This indicated that the three genes were positive regulators of anthocyanin synthesis. Their upstream genes were correspondingly up-regulated, such as *C4H*, *4CL*, *CHS*, *CHI*, and *F3H* (Figure 8G). The quantitative data proved that three genes were the key factors in regulating anthocyanin accumulation through the cyanidin pathway in the floral buds of lilies. It also laid a foundation for us to analyze the relationship between anthocyanins and vernalization in the future.

## 5. Conclusions

In this study, it was revealed that the maturation of lily bulbs is closely associated with the amount of anthocyanin accumulation. Therefore, anthocyanin accumulation in flower buds is a very important marker in the development of lily bulbs, and the concentration of anthocyanins directly decides on the ripeness of bulbs. In addition, the maturity of lily bulbs will also directly decide on the time of bulb quality and lily vernalization. The transcriptomics and VIGS analysis firstly revealed that *LhMYB114*, *LhANS-rr1,* and *LhDFR* were the key genes for regulating the accumulation of cyanidin in floral lily buds. Among them, *LhMYB114* is an important regulatory transcription factor during lily bulb maturation. In a word, the data in our article contribute to the understanding for the molecular mechanism of anthocyanin biosynthesis and the judging of bulb maturity in *Lilium* flower bud development.

## Figures and Tables

**Figure 1 genes-14-00559-f001:**
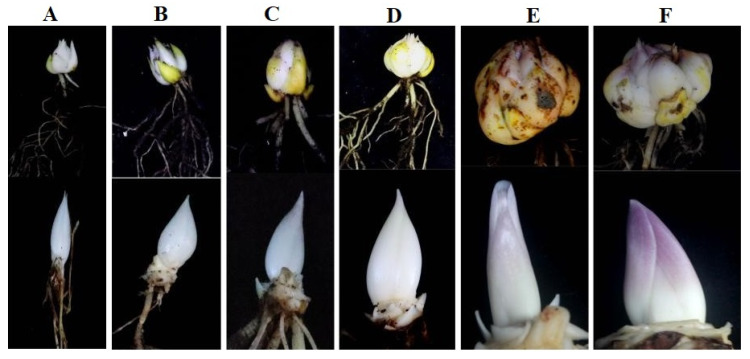
The appearance and morphology of *Lilium* bulb and bud at different developmental stages (**A**–**F**). (**A**) The bulb diameter is 3 cm; (**B**) the bulb diameter is 6 cm; (**C**) the bulb diameter is 10 cm; (**D**) the bulb diameter is 14 cm; (**E**) the bulb diameter is 16 cm; (**F**) the bulb diameter is 22 cm. The data represent the average of three biological replicates.

**Figure 2 genes-14-00559-f002:**
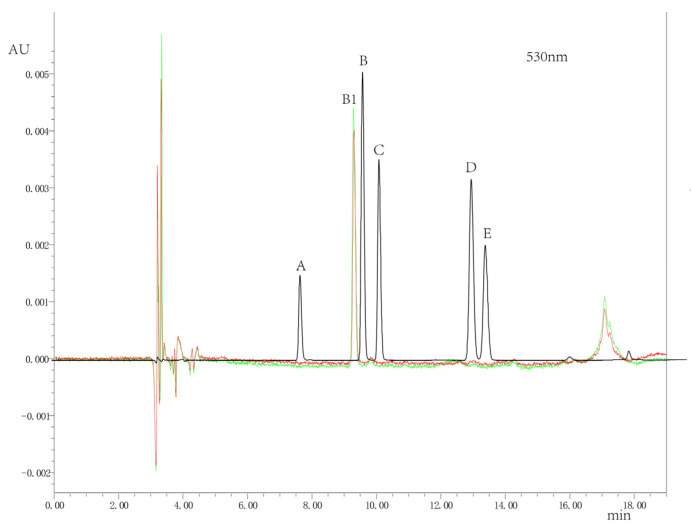
The concentrations of anthocyanin-related compounds in the flower buds of oriental lily ‘Siberia’. High-performance liquid chromatography-mass spectrometry (HPLC-MS) chromatograms suggested the presence of anthocyanin aglycones was completed at 530nm; the compounds were identified by comparing with the mass spectra of the standards. A—delphinidin, B/B1—cyanidin, C—petunidin, D—pelargonin, E—mallow pigment. Black color represents the standard, and red or green represents the text sample.

**Figure 3 genes-14-00559-f003:**
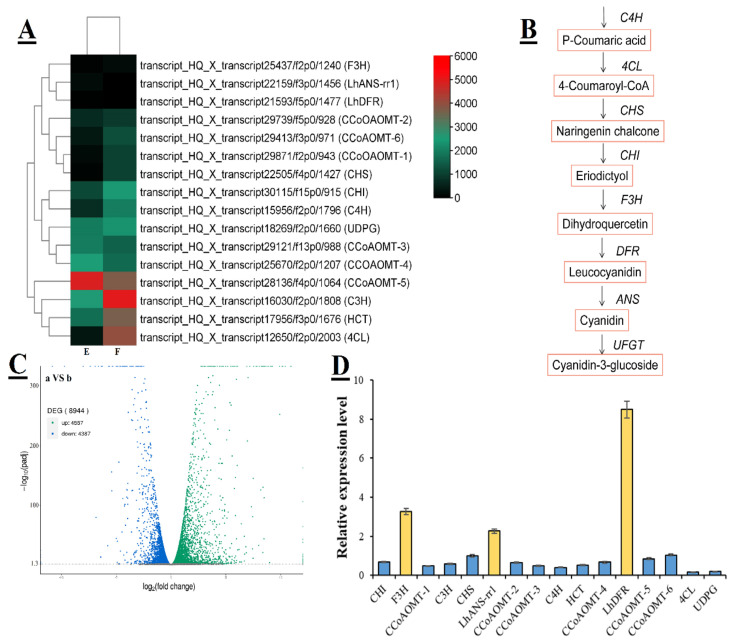
Transcriptome analysis and qRT-PCR validation of the candidate DEGs involved in anthocyanin biosynthesis in *Lilium* ′Siberia′. (**A**) A heatmap showing the expression patterns of the DEGs involved in anthocyanin biosynthesis. The heatmap was generated based on the normalized log^2^ (FoldChange) > 1 values for each DEG. Green and red scales represent relatively low or high expression, respectively. (**B**) Schematic diagram of metabolic pathway for cyanidin. (**C**) Volcano plot of the total differentially expressed genes in both E and F groups using the threshold of *p* < 0.05 and |log2FoldChange| > 1. The *x*-axis represents the log_2_ (fold change) values for gene expression, and the *y*-axis represents the −log_10_ (*p* value). (**D**) The qRT–PCR validation of the DEGs involved in anthocyanin biosynthesis. Expression data are the represent as the mean values of three biological replicates ± SD. Yellow color represents significant up-regulation of genes, while blue color represents insignificant changes.

**Figure 4 genes-14-00559-f004:**
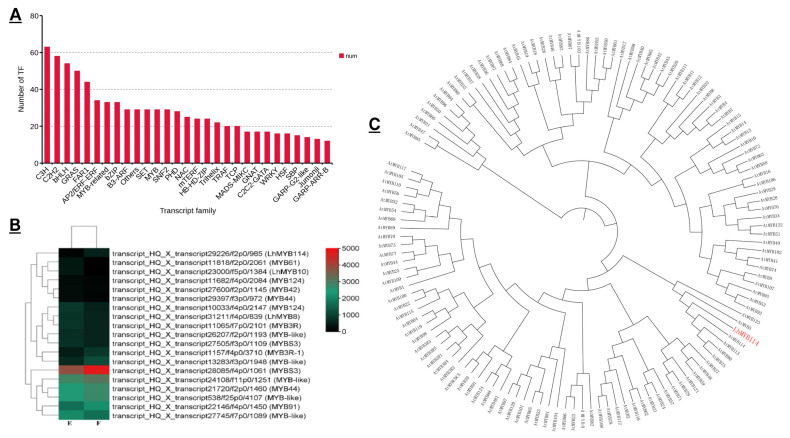
Genetic expressions of *MYB* genes and cluster analysis in transcriptome data. (**A**) The numbers and families of top 28 TFs enriched in flower bud. (**B**) The heatmap analysis of *MYB* DEGs through transcriptome data. The heatmap was generated based on the normalized log^2^ (FoldChange) > 1 values for each DEG. The color scale at the right represents gene expression values (the red corresponds to genes with high expression and the green corresponds to genes with low expression). (**C**) The cluster analysis map of R2R3-MYB proteins. The cluster analysis tree was completed using MEGA 5.0 software with the *Arabidopsis MYB* genes as reference. The aim gene of *LhMYB114* is highlighted in red color.

**Figure 5 genes-14-00559-f005:**
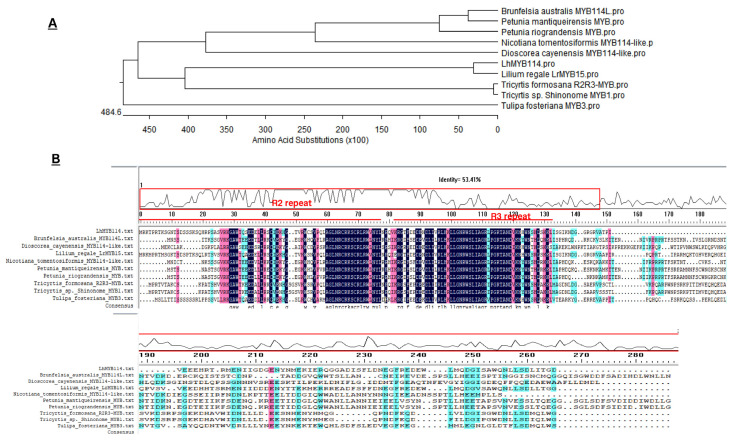
Amino acid sequence alignment of *LhMYB114*. (**A**) The phylogenetic analysis of *LhMYB114* with homologous genes related to anthocyanin biosynthesis. (**B**) The important domain analysis of *LhMYB114*. Color represents amino acid sequence is high similarity, including pink, blue, cyan.

**Figure 6 genes-14-00559-f006:**
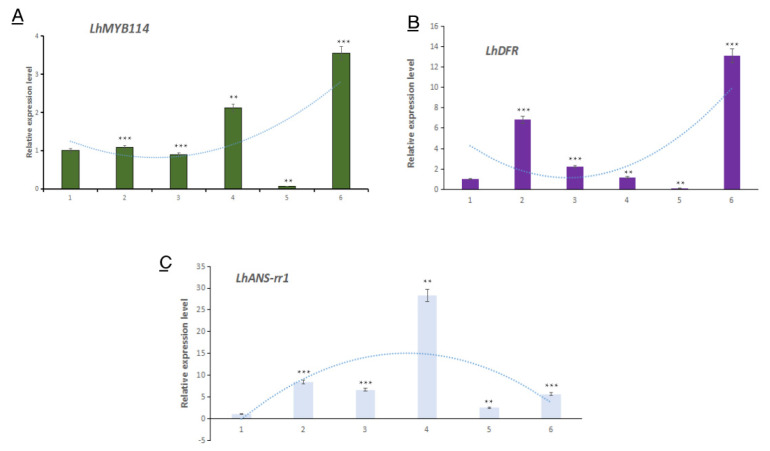
The qRT–PCR analysis of 3 key genes responsible for anthocyanin metabolism in flower buds of lily at different development stages. (**A**) The expression of *LhMYB114* genes at different stages. (**B**) The expression of *LhDFR* genes at different stages. (**C**) The expression of *LhANS-rr1* genes at different stages. The values are means ± SDs (*n* = 6). The data were normalized to a value of 1 for the A group. Significant differences were determined by Student′s *t* test (** *p* < 0.01; *** *p* < 0.001).

**Figure 7 genes-14-00559-f007:**
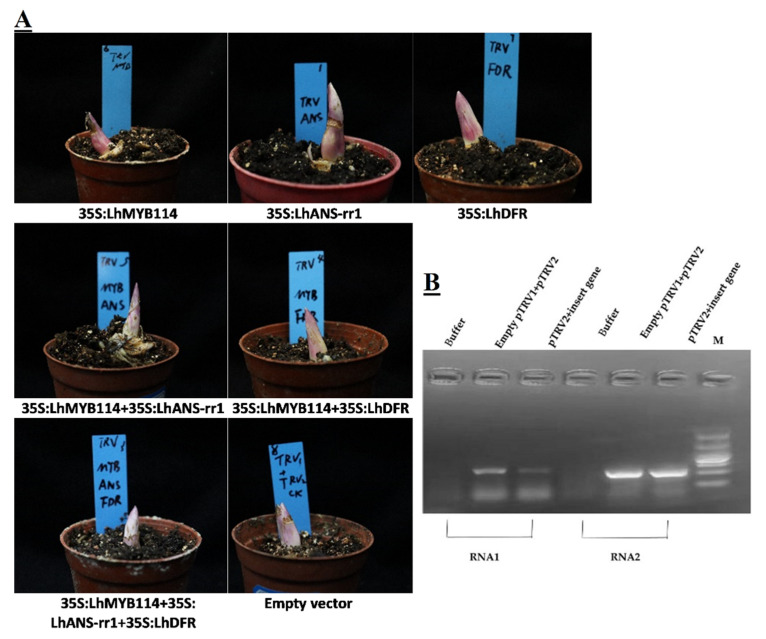
Transient silencing phenotypes among *LhMYB114*, *LhANS-rr1,* and *LhDFR* in *Lilium* flower buds by virus-induced gene silencing (VIGS). (**A**) Photographs of *Lilium* flower buds from TRV-infected plants. (**B**) RT-PCR detection of RNA1 and RNA2 of TRV1 and TRV2 in *Lilium*.

**Figure 8 genes-14-00559-f008:**
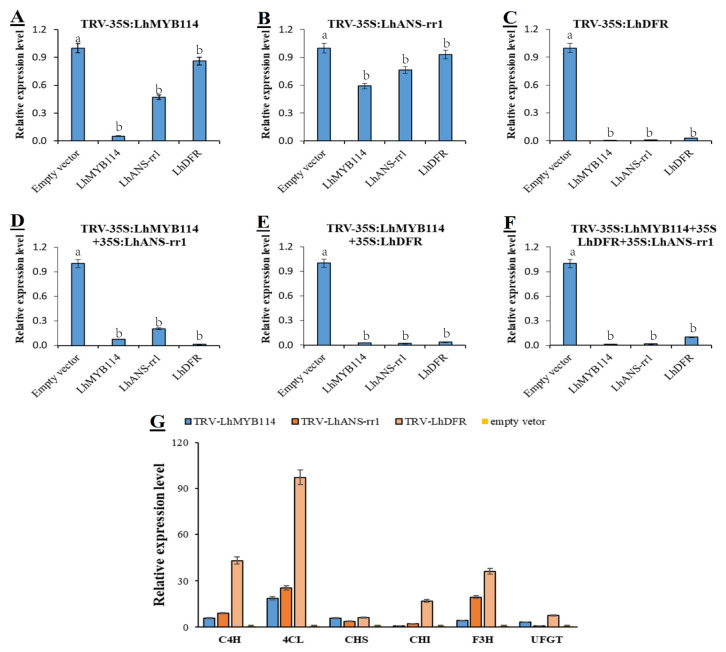
Relative expression levels of three key genes under different VIGS treatments. (**A**) Down-regulation of *LhANS-rr1* and *LhDFR* expression in *LhMYB114*-silenced bud. (**B**) Down-regulation of *LhMYB114* and *LhDFR* expression in *LhANS-rr1*-silenced bud. (**C**) Down-regulation of *LhANS-rr1* and *LhMYB114* expression in *LhDFR*-silenced bud. (**D**) Down-regulation of *LhMYB114*, *LhANS-rr1* and *LhDFR* expression in *LhMYB114*+*LhANS-rr1*-silenced bud. (**E**) Down-regulation of *LhMYB114*, *LhANS-rr1* and *LhDFR* expression in *LhMYB114*+*LhDFR*-silenced bud. (**F**) Down-regulation of *LhMYB114*, *LhANS-rr1* and *LhDFR* expression in *LhMYB114*+*LhDFR*+*LhANS-rr1*-silenced bud. (**G**) The six main genes (*C4H*, *4CL*, *CHS*, *CHI*, *F3H,* and *UFGT*) expressed in TRV-*LhMYB114*, TRV-*LhANS-rr1*, TRV-*LhDFR*, and empty vector. The values are means ± SDs (*n* = 3). Lowercase letters (a,b) indicate statistically significant differences at *p* < 0.05.

## Data Availability

Not applicable.

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
