# Peer review of "LhANS-rr1*, *LhDFR,* and *LhMYB114* Regulate Anthocyanin Biosynthesis in Flower Buds of *Lilium* ‘Siberia’"

_genes, 2023, doi:10.3390/genes14030559_

Round 1

Reviewer 1 Report

Review of genes-2086567

 LhANS-rr1LhDFR and LhMYB114 Regulate Anthocyanin Biosynthesis in Flower Bud of Oriental Lily ‘Siberia’

Shaozhong Fang, Mi Lin, Muhammad Moaaz Ali, Yiping Zheng, Xiaoyan Yi, Shaojuan Wang, Faxing Chen, Zhimin Lin

The authors wished to study the regulation of anthocyanin synthesis in lily bulbs. They therefore measured the types and amounts of anthocyanins present at various stages of lily development, and performed transcriptomic analysis of two later stages in development that showed the greatest changes in anthocyanin content. This analysis identified two genes related to anthocyanin synthesis that were upregulated at these stages. They also identified a transcription factorLhMYB114 that was directly involved in anthocyanin accumulation. Results of the transcriptomic analysis were validated by RT-qPCR assays of the genes they identified, and they used virus-induced gene silencing (VIGS) to show that down-regulation of LhANS-rr1LhDFR, and LhMYB114 decreased anthocyanin accumulation in the treated tissues.  The authors concluded that they identified suitable targets for manipulating lily flower color and provided new insights into lily floral development.

Overall, the study seems to have been performed competently using suitable techniques, but a major concern is that in several experiments it is unclear how many replicates were performed and the statistical analysis used.  For example, how many replicates were performed for the anthocyanin analysis?  This should be indicated in the relevant sections of the Materials and Methods and in the captions of the relevant figures. In contrast, they did a good job of indicating the number of replicates in section 2.4.  However, this information should also be provided in the captions to the relevant figures. 

Captions to figures that show error bars should indicate the number of replicates, whether the error bars represent Standard Deviation (SD) or Standard error (SE) if some differences are significant, the statistical test used and the level of significance. Statistical analysis should also be described in the materials and methods.

The methods used for anthocyanidin identification should be listed in section 2.2, including the standards used and that the compounds were identified based on retention times and mass spectra. Why focus just on the major anthocyanidin?  Might not other anthocyanidins contribute to the flower color?

The bioinformatics pipeline used for processing transcriptomic data and identifying DEGs should be described in the materials and methods, either as part of section 2.3 or in its own section. They should also describe how KEGG enrichment analysis was performed.

It seems that vernalization is important for the development of lily flower color.  This must be explained in the introduction and again in the results and discussion, since many readers go directly to the results to see if there is anything reported that is interesting to them.

Overall, the English is good, but there are many sentences that need correction.  Most are relatively minor. However, some of them make it hard to understand the meaning. I therefore recommend editing by a native English speaker.

Here are some examples of sentences that need correction, but there are many more.

Please list the latin binomial for oriental lily in the title.

Lines 17-18: is cyanidin the major anthocyanin found in lily, or in this variety of lily?

Lines 19-24 are hard to understand and should be rewritten for clarity.

Lines 50-52 are hard to understand and should be rewritten for clarity. Do you mean that if GST proteins are not present then anthocyanins cannot be delivered?

Lines 53-57 are hard to understand and should be rewritten for clarity.

Lines 88-91 need more explanation. Was the difference in anthocyanins the reason that they chose to the last 2 stages for transcriptomics analysis? They should also explain why they wished to study the regulation of anthocyanin production via transcriptomics.

Line 97: what is fresh sample powder?

Lines 97-100: as stated above, this section must provide additional information and must also be rewritten for clarity.

Section 2.4: what kit was used for the RT-qPCR analysis? What statistical procedures were used to identify DEG?

Line 121: please explain better how the fragments specific to the target gene were identified.

Section 2.5 must be rewritten for clarity. Please clarify that the agrobacterium strains contain plasmids encoding the indicated genes. Also please explain what “pinhead infiltration” is.

Section 2.6 should provide more information. How were the samples stained for SEM?

Lines 141-143 are M&M and not “results”

Caption to fig 2: how many samples were analyzed? How was significance of results determined? Please explain in the caption which peaks correspond to the plant extracts.

Line 154: explain better how you identified 16 DEGS.

Lines 153-159 must be rewritten for clarity. Also please explain that these genes encode enzymes that catalyze anthocyanin synthesis.

Caption to fig 3: how many samples were analyzed? How was significance of results determined? What was the reference gene?

Lines 169-172 must be rewritten for clarity.

Lines 172-173: how did you select these genes for analysis?

Caption to fig 4: explain better!  How was the heatmap performed? How was the cluster analysis performed? What genes were used?

Lines 184-188 must be rewritten for clarity.

Lines 196-198 must be rewritten to explain why you chose these genes for analysis.

Caption to fig 6: explain better!  What do A-F represent? What was the internal control? How many replicates? Error bars?

Lines 204-208 must be rewritten for clarity. You first incubated 7 days at 25 ˚C in the dark, then switched to 7-8 °C ice storage to continue germination? This doesn’t make sense to most plant biologists. 

Line 210: what are RNA1 and RNA2?

Line 213: TRV didn’t invade, Agrobacteria harboring the plasmid encoding TRV were what invaded.

Caption to fig 8: explain better!  What was the internal control? How many replicates? Error bars? Statistical analysis?

Lines 253-255 must be rewritten for clarity.

Lines 262-267 must be rewritten for clarity.

Lines 270-272 must be rewritten for clarity.

Lines 274-281 must be rewritten for clarity.

Lines 285-299 must be rewritten for clarity. 

Reviewer 2 Report

Manuscript "LhANS-rr1, LhDFR and LhMYB114 Regulate Anthocyanin Biosynthesis in Flower Bud of Oriental Lily ‘Siberia’" is very interesting.

General comments:

Authors isolated LhMYB114 transcription factor from Oriental hybrid Lilium 'Siberia' based on the transcriptome data. These study suggested that it can effectively regulate the genetic expression of ANS and DFR by the VIGS system, thereby affecting the accumulation of anthocyanins in the floral bud of the Lilium bulb.
Authors analysed the floral buds of Lilium hybrid 'Siberia' at six development stages.

Detailed comments:
The description of the research material is correct. Unfortunately, the subsection "Statistical analysis" is missing.
The manuscript lacks statistical analysis of the data taken.
Figure 6 shows the regression curves. Their description and significance testing of the parameters of the obtained models are missing.

My suggestion:
The work should be improved. Conduct statistical analysis on, among other things, the comparison of the six stages.

Paper needs major revision.

Reviewer 3 Report

In this article, the authors present investigations of genes regulating anthocyanin biosynthesis. The variety 'Siberia', which belongs to the oriental lily group, was selected as the object of research. The lily hybrids of this group are highly valued in horticulture due to their beautiful and aromatic flowers. Thus, the conducted research is relevant both in the scientific-fundamental and practical sense. However, the content of the submitted article must be significantly corrected.

 Main comments

Title and keywords are proper.

Abstract. The last sentence should be clarified. Can the conducted research become the basis for further discussions about bulb development or anthocyanin biosynthesis?

Introduction. This section should deal with the problem raised, discuss the relevance and novelty of the presented research. I missed an essential point in the introduction - the main goal was not formulated. The last paragraph should be moved to the discussion or conclusion section.

Materials and methods. There is a lack of definition when examining Figure 1. The different developmental stages (A, B...) are not explained. On the other hand, the description of figures must be very specific and clear without the main text. Please specify the identification of anthocyanins: extract preparation, equipment (brand, country), etc. It is necessary to provide the information on which statistical analysis methods were used.

Results. Please clarify the terms in Figure 2. and in the text. In my opinion that the terms anthocyanins and anthocyanidins were confused. Delfinidin is an anthocyanidin, pelargonin is an anthocyanin etc. What does it mean „petunia“ (line 150)?

Did you determined the concentration of anthocyanins? (Figure 2).

Please provide statistical reasoning of data in Figure 8.

Did you investigate „anthocyanins biosynthesis“ or „anthocyanins accumulation“? (lines 203, 298).

Discussion. Please clarify the terms in this section as well (lines 264-265).

Conclusions. This section is non-informative and the last sentence is simply a declarative phrase. Please accurately and briefly discuss the essence of the results obtained.

 Minor remarks

Change „bub“ with „bulb“ (line 28).

Change „ antioxidants“ with „metabolites“.

Latin names of plants must be written in italics throughout the article. On the other hand, when the name of a genus or species of a plant is mentioned for the first time, the initials of the authors must be indicated, e.g. Hydrangea macrophylla (Thunb.) Ser.

You can find plant names here: to http://www.worldfloraonline.org/

Round 2

Reviewer 1 Report

Review of revised genes-2086567

 LhANS-rr1, LhDFR and LhMYB114 Regulate Anthocyanin Biosynthesis in Flower Bud of Oriental Lily ‘Siberia’

Shaozhong Fang, Mi Lin, Muhammad Moaaz Ali, Yiping Zheng, Xiaoyan Yi, Shaojuan Wang, Faxing Chen, Zhimin Lin

The authors wished to study the regulation of anthocyanin synthesis in lily bulbs. They therefore measured the types and amounts of anthocyanins present at various stages of lily development, and performed transcriptomic analysis of two later stages in development that showed the greatest changes in anthocyanin content. This analysis identified two genes related to anthocyanin synthesis that were upregulated at these stages. They also identified a transcription factor LhMYB114 that was directly involved in anthocyanin accumulation. Results of the transcriptomic analysis were validated by RT-qPCR assays of the genes they identified, and they used virus-induced gene silencing (VIGS) to show that down-regulation of LhANS-rr1, LhDFR, and LhMYB114 decreased anthocyanin accumulation in the treated tissues.  The authors concluded that they identified suitable targets for manipulating lily flower color and provided new insights into lily floral development.

The authors have addressed many of the concerns raised in my first review, but there still remain many issues.

Perhaps the most fundamental issue is that “Genes” is directed at a readership that is interested in genes but may not know much about plants.  Therefore, the authors need to provide more information about their organism and the reasons for the treatments they provided. It is imply unreasonable to assume that the average reader of “Genes” will know that you first needed to keep lilies in the dark at 25˚ C for Agrobacterium growth, but then keep them at 7-8˚C to induce germination. This must be explained better!

The authors also need to state the numbers of biological and technical replicates used for figures 6 and 8, the reference gene, and the statisitical test used to determine significance. Without this information the pare can not be published.

The authors also need to explain that Agrobacterium deliver the TRV sequences.  TRV don’t grow.  Agrobacteria delivering TRV are what grow.

It seems that vernalization is important for the development of lily flower color.  This must be explained in the introduction and again in the results and discussion, since many readers go directly to the results to see if there is anything reported that is interesting to them.

Lines 101-106 are hard to understand and must be rewritten for clarity and to correct mistakes. They must also either state the composition of the extraction solution or provide a reference.

Lines 122-124 are hard to understand and must be rewritten.

Lines 144-146 must be rewritten to explain what pinhead infiltration is and how it is performed.  This is not a technique that even most plant biologists are familiar with, much less readers working on other groups of organisms.

Lines 156-157 and figure 2 are results, not Materials and Methods, and must be moved to the results section.

Lines 181-182: how many samples were analyzed? How was significance of results determined? What was the reference gene? What do the error bars represent?

Lines 169-172 must be rewritten for clarity.

Lines 172-173: how did you select these genes for analysis?

Caption to fig 4: explain better!  How was the heatmap performed? How was the cluster analysis performed? What genes were used?os was the clutsH

Lines 184-191 must be rewritten for clarity.

Lines 199-200 don’t make sense and must be rewritten.

Caption to fig 6 must state how many biological and how many technical replicates were performed, must include error bars, must explain what test was used to determine statistical significance and at what level and what the reference gene is. As presented it is unsuitable for publication. os was the clutsH

Lines 219- 232  must be rewritten for clarity and to correct mistakes.TRV didn’t invade, Agrobacteria harboring the plasmids encoding TRV1 and TRV2 were what invaded. They must explain what RNA 1 and RNA 2 are, even most plant geneticists much less Drosophila or yeast geneticists will understand what they are talking about.

Caption to fig 8 must state how many biological and how many technical replicates were performed, must include error bars, must explain what test was used to determine statistical significance and at what level and what the reference gene is. As presented it is unsuitable for publication. os was the clutsH

Lines 278-280 must be rewritten for clarity.

Lines 319-321 must be rewritten for clarity.

Reviewer 2 Report

Unfortunately, still (!) the subsection "Statistical analysis" is missing.
The manuscript still (!) lacks statistical analysis of the data taken.
Figure 6 shows the regression curves. Their description and significance testing of the parameters of the obtained models still (!) are missing.
The work should be improved. Conduct statistical analysis on, among other things, the comparison of the six stages.

Reviewer 3 Report

The authors corrected the manuscript based on suggestions and comments.
